# Edge effects on tree architecture exacerbate biomass loss of fragmented Amazonian forests

Matheus Henrique Nunes [1,2] ✉, Marcel Caritá Vaz[3], José Luís Campana Camargo [4,5], William F. Laurance [6], Ana de Andrade [5], Alberto Vicentini[5,7], Susan Laurance [6], Pasi Raumonen [8], Toby Jackson [9], Gabriela Zuquim[10], Jin Wu [11], Josep Peñuelas [12,13], Jérôme Chave [14] & Eduardo Eiji Maeda [1,15] ✉

Habitat fragmentation could potentially affect tree architecture and allometry. Here, we use ground surveys of terrestrial LiDAR in Central Amazonia to explore the influence of forest edge effects on tree architecture and allometry, as well as forest biomass, 40 years after fragmentation. We find that young trees colonising the forest fragments have thicker branches and architectural traits that optimise for light capture, which result in 50% more woody volume than their counterparts of similar stem size and height in the forest interior. However, we observe a disproportionately lower height in some large trees, leading to a 30% decline in their woody volume. Despite the substantial wood production of colonising trees, the lower height of some large trees has resulted in a net loss of 6.0 Mg ha$^{-1}$ of aboveground biomass – representing 2.3% of the aboveground biomass of edge forests. Our findings indicate a strong influence of edge effects on tree architecture and allometry, and uncover an overlooked factor that likely exacerbates carbon losses in fragmented forests.

The three-dimensional form of trees, or tree architecture, reflects the allocation of photosynthetically fixed carbon within the plants. Tree architecture can be considered a by-product of environmental pressures on plant growth, reproduction and survival[1,2]. Fine adjustments of the aboveground architecture of trees can minimise competition from neighbouring trees, improve hydraulic conductance, limit transpiration and maximise light capture[3–8]. In Amazonian forests, trees vary greatly in size and architecture across species, as a result of evolutionary processes over millions of years[9–11]. Size and architecture also vary considerably across individuals due to short- to mid-term acclimation and adaptation to changing environmental conditions, including canopy gaps caused by the mortality of large trees and forest

[1]Department of Geosciences and Geography, University of Helsinki, Helsinki, Finland. [2]Department of Geographical Sciences, University of Maryland, College Park, MD, USA. [3]Institute for Environmental Science and Sustainabilty, Wilkes University, Wilkes-Barre, PA, USA. [4]Ecology Graduate Program, National Institute for Amazonian Research, (INPA), Manaus, Brazil. [5]Biological Dynamics of Forest Fragments Project (BDFFP) at National Institute for Amazonian Research (INPA), Manaus, Brazil. [6]Centre for Tropical Environmental and Sustainability Science, College of Science and Engineering, James Cook University, Cairns, Queensland, Australia. [7]Coordenação de Pesquisas em Ecologia, Instituto Nacional de Pesquisas da Amazônia (INPA), Manaus, AM, Brasil. [8]Computing Sciences, Tampere University, Tampere, Finland. [9]Plant Sciences and Conservation Research Institute, University of Cambridge, Cambridge, United Kingdom. [10]Amazon Research Team, Department of Biology, University of Turku, Turku, Finland. [11]School of Biological Sciences and Institute for Climate and Carbon Neutrality, The University of Hong Kong, Hong Kong, China. [12]CREAF, Cerdanyola del Vallès, Barcelona, Catalonia, Spain. [13]CSIC, Global Ecology Unit CREAF-CSIC-UAB, Bellaterra, Barcelona, Catalonia, Spain. [14]Laboratoire Evolution et Diversité Biologique, CNRS, UPS, IRD, Université Paul Sabatier, Toulouse, France. [15]Finnish Meteorological Institute, FMI, Helsinki, Finland. ✉e-mail: mhnunes@umd.edu; eduardo.maeda@helsinki.fi

blowdowns[6,12–15]. The architectural traits of Amazonian trees control $CO_2$ loss from stem and branch respiration, hydraulic safety and efficiency, light capture and mechanical stability, which together modulate biomass allocation and carbon storage[16–19]. Changes in tree architecture could therefore reveal biome-wide impacts on carbon cycling, with regional and global influences on vegetation feedbacks[1,9,20,21].

The architecture of Amazonian trees could be affected by disturbances arising from forest fragmentation. The edges of forest fragments tend to have greater light availability due to the mortality of large trees and lateral light penetrating from the edges[22–24]. This may induce changes in tree architecture to optimise the capture and use of light under these new circumstances, including higher vertical and horizontal crown growth that modify branching patterns and crown shape[17,25,26]. Higher temperatures and lower water availability in forest edges increase the evaporative demand of the vegetation[27,28], and trees can shorten the distances for transporting water and nutrients to minimise hydraulic conductance[7]. High wind turbulence near the fragment edges may kill highly asymmetrical trees that deviate their centre of gravity substantially from their stems[29,30]. On the other hand, highly symmetrical trees can have greater mechanical stability but a limited ability to avoid competition for light with neighbouring trees that can fit their crowns into irregular spaces. To complicate matters further, the high mortality of large individuals can damage neighbouring trees with suppressed aboveground biomass allocation, which can have large effects on their architecture and size[31]. However, large uncertainties remain regarding the effects of forest fragmentation on tree architecture, particularly because (i) tree architecture varies considerably across life stages[32] (ii) multiple architectural attributes interact to affect the structural complexity of individual trees[2], and (iii) the responses of trees to forest fragmentation vary enormously within and between species[33].

Edge effects on tree architecture could affect allometric models that predict the aboveground biomass (AGB) of fragmented Amazonian forests as a function of more easily measurable properties of stem diameter (DBH, diameter at breast height) and tree height[34]. Long-term tree measurements have shown that forest fragments in Central Amazonia experience a dramatic loss of aboveground tree biomass caused by the mortality of large trees that is not offset by the growth and recruitment of new trees[22,23]. However, differences in tree allometry caused by edge effects on tree architecture could either lead to

additional losses in AGB (i.e. thinner and shorter branches, crown damages) or offset this biomass loss in fragmented forests. Terrestrial laser scanning (TLS) or "terrestrial LiDAR" surveys can be of particular importance to reduce uncertainties in tree volume estimates by considering the geometry and shape of trees, and without the difficulties associated with traditional destructive methods of tree measurement[34]. Furthermore, TLS surveys offer new perspectives into the three-dimensional (3D) structure of trees (Fig. 1), including fine-scale architectural traits[2,35] and accurate tree allometry estimates[36]. Understanding how forest edge trees adjust their architecture and allometry can help us predict how plants acclimate to and survive environmental changes, and their impacts on biogeochemical fluxes and on the terrestrial carbon cycle.

Here, we tested the hypotheses that: (1) both pre-existing trees established before forest fragmentation and trees that colonised the forest fragments had their architectural traits and allometry affected by forest edges, given the higher light availability[24], higher wind turbulence[29], the hotter and drier conditions near these forest edges[28] and the high mortality of large trees that may damage their neighbours;[23] and (2) the AGB of fragmented forests is impacted by edge effects on tree allometry, with potentially significant biome-wide implications. To address these questions, we investigate the long-term edge effects on tree architecture and tree allometry 40 years after forest fragmentation, and how changes in tree allometry impact the AGB of fragmented Amazonian forests. To quantify the edge effects on tree architecture, we conducted fieldwork in the Biological Dynamics of Forest Fragments Project (BDFFP) in Central Amazonia, the world's longest-running experimental study of habitat fragmentation[37]. We used TLS surveys to measure the 3D structure of trees and develop allometric models for edge versus interior trees. We then used more than 12,000 DBH measurements to quantify the impacts of fragmentation-related allometric changes on the forest AGB. We find that edge effects impact the allometry of trees by altering the architecture of both colonising and surviving trees, and these effects exacerbate the biomass loss of fragmented Amazonian forests.

## Results
### Edge effects on architectural traits
We used tree height to determine whether trees had been established before or after forest fragmentation. Trees above 20 m in height have

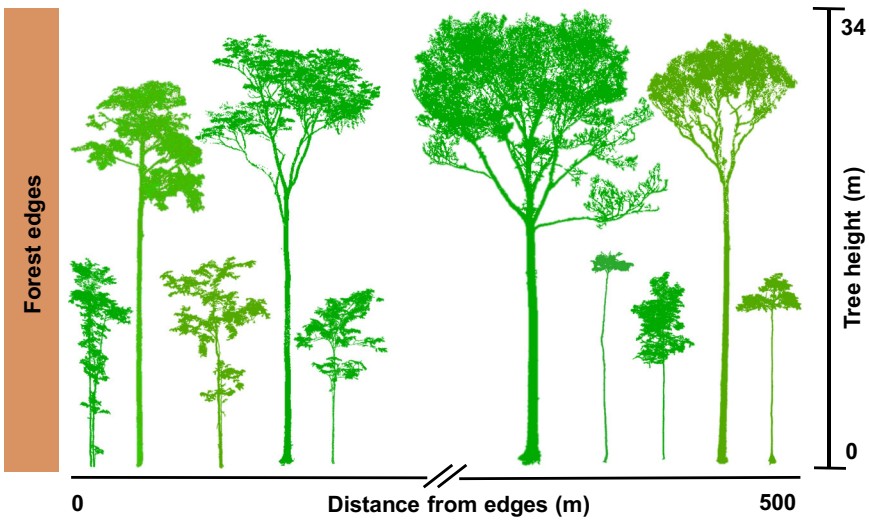

**Fig. 1 | Point cloud of trees growing in fragmented forests in Central Amazonia, obtained using high-resolution terrestrial laser scanning (TLS).** The quantification of architectural attributes of tropical trees is challenging and has been largely overlooked[10]. TLS offers new perspectives into the three-dimensional (3D) structure of trees, including descriptions of fine-scale architectural traits such as tree asymmetry and vertical distribution of branches. Here, we investigate how the architecture of surviving and young colonising trees change with proximity to forest edges. We used TLS data that resulted in a point spacing of 1.4 cm at a 20 m distance from the scanner.

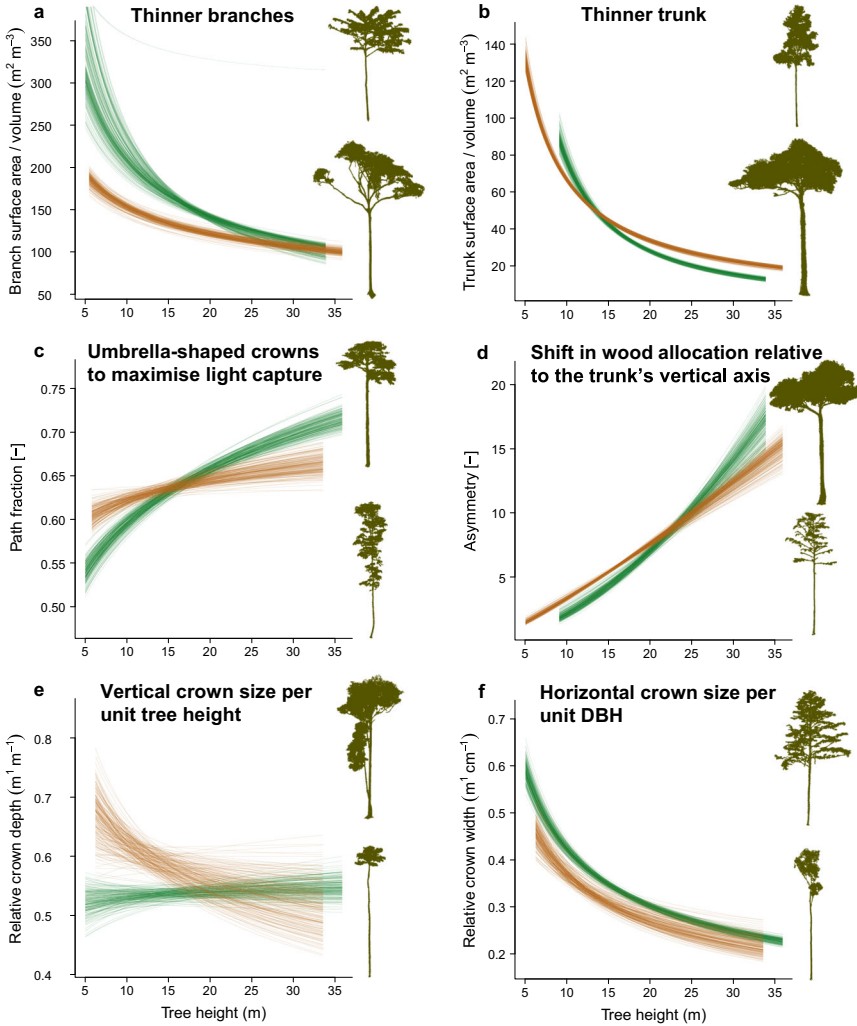

**Fig. 2 | Architectural traits acquired using a Terrestrial Laser Scanner (TLS) in Central Amazonia varied with tree height and were affected by edge effects.** Nonlinear mixed-effects models were used to predict **a** surface area per unit volume (m² m⁻³) of branches, **b** surface area per unit volume (m² m⁻³) of trunks, **c** path fraction, **d** asymmetry, **e** relative crown depth (m m⁻¹), and **f** relative crown width (m cm⁻¹) of trees in forest edges (orange) and forest interior (green). Each line corresponds to the model prediction obtained by fitting 200 randomised permutations of subsets split into 80/20 for calibration and validation, respectively. Plot titles indicate the ecological meaning of high trait values. TLS-based trees are shown for a visual representation of the variation between low and high trait values.

been established before the forest fragmentation in 1980 and thus have survived the 40-year-old edge effects. Two-thirds of trees below 20 m in height were post-fragmentation recruits, which demonstrates that a large proportion of small trees have already been exposed to edge effects during their juvenile phase (Supplementary Fig. 3). Edge effects on architectural traits varied in extent; differences in relative crown width and relative crown depth were most pronounced within 10 m from the edges, ~20 m for path fraction, ~40 m for trunk and branch surface area unit volume and 55 m for asymmetry (Supplementary Fig. 4).

Our models demonstrated that edge effects affected architectural traits, but these effects were dependent on when plants were established in the forest fragments (Fig. 2). The surviving tall trees in the edges had higher surface area per unit volume of trunks ($CI_{95\%}$: 24–26 m² m⁻³ in the edge vs. $CI_{95\%}$: 14–16 m² m⁻³ in the interior), which demonstrates that edge effects led to thinner trunks (as thinner objects have a higher area per unit volume). Tall trees were more symmetrical in the edges than in the interior forests ($CI_{95\%}$: 14–18 in the edge vs. 11–13 in the interior) and had a reduced path fraction ($CI_{95\%}$: 0.63 − 0.67 in the edge vs. 0.69−0.73 in the interior). Relative crown width ($CI_{95\%}$: 0.20−0.29 m cm⁻¹ in the edges vs. 0.25−0.27 m cm⁻¹ in the interior) and relative crown depth

($CI_{95\%}$: 0.45−0.65 m m⁻¹ in the edges vs. 0.51−0.56 m m⁻¹ in the interior) of tall trees in the edges were not distinct from those of interior forests, indicating that crown dimensions in the edges were proportional to trunk size and tree height compared to trees in the interior.

Our results also demonstrate that short trees colonising the forest edges had thicker branches and trunks, owing to reduced branch surface area per unit volume ($CI_{95\%}$: 170 − 200 m² m⁻³ in the edges vs. 250−400 m² m⁻³ in the interior) and trunk surface area per unit volume ($CI_{95\%}$: 74−77 m² m⁻³ in the edges vs. 80−100 m² m⁻³ in the interior), trees were more asymmetrical ($CI_{95\%}$: 3.0−3.2 in the edges vs. 2.0−2.5 in the interior) and had higher path fraction ($CI_{95\%}$: 0.59−0.63 in the edges vs. 0.53−0.57 in the interior). The crowns of small trees in the edges had larger relative depth most possibly owing to multi-stemmed trees that colonised the forest edges ($CI_{95\%}$: 0.60−0.75 m m⁻¹ in the edges vs. 0.50−0.55 m m⁻¹ in the interior), but smaller relative width given the large trunk sizes of short trees in the forest edges ($CI_{95\%}$: 0.41−0.50 m cm⁻¹ in the edges vs. 0.51−0.56 m cm⁻¹ in the interior).

The within-plot variability - seen as local effects arising from edge effects, as well as species-specific and ontogenetic influences - accounted for most of the trait variability (Supplementary Figure 5).

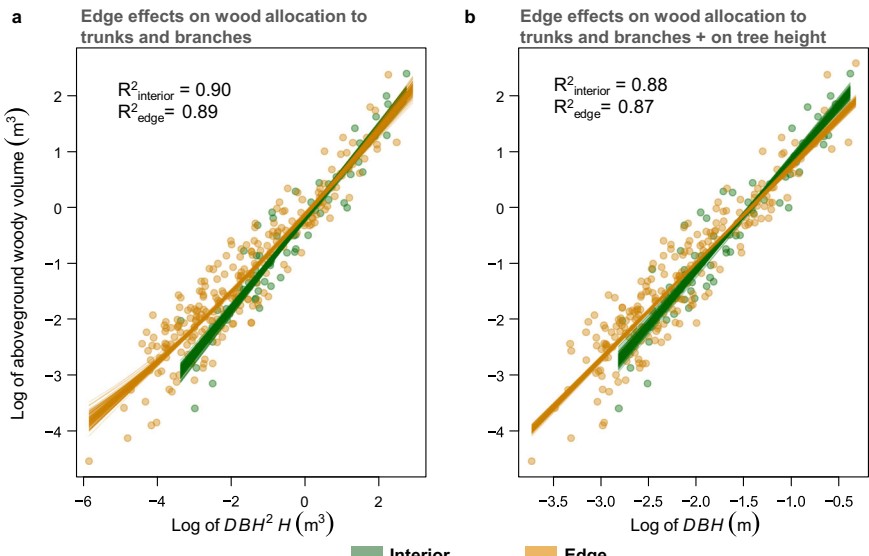

**a** Edge effects on wood allocation to trunks and branches

$R^2_{interior} = 0.90$
$R^2_{edge} = 0.89$

**b** Edge effects on wood allocation to trunks and branches + on tree height

$R^2_{interior} = 0.88$
$R^2_{edge} = 0.87$

Interior   Edge

**Fig. 3 | Allometric models for predicting the woody volume of Amazonian trees.** DBH is the diameter at breast height measured at 1.3 m above the ground in cm, and H is the total tree height in metres. Tree measurements were acquired using a Terrestrial Laser Scanner (TLS) within the Biological Dynamics of Forest Fragments Project (BDFFP) in Central Amazonia. Plot identity nested within landscape (position of fragment within the landscape and fragment size) and region within Central Amazonia were included as random variables. Points represent the observed values and each line corresponds to the model prediction obtained by fitting 200 randomised permutations of subsets split into 80/20 for calibration and validation, respectively.

The within-plot variability also includes analytical errors, such as those arising from measurement, co-registration, tree extraction and QSM – although we were unable to quantify them.

Architectural traits, except relative crown depth, were correlated with the woody volume of individual trees (Supplementary Figure 6). In particular, one component of variation was tightly linked to variation in woody volume, and explained nearly half of all the changes in architectural traits (Supplementary methods 5; Supplementary Figure 7). The component shows that trees with thicker branches and trunks had higher woody volume, but they tended to be more asymmetrical and had a higher path fraction.

### Edge effects on tree allometry

We developed allometric equations for trees in the forest interior and forest edges that predict the aboveground woody volume as a function of stem size and tree height (DBH² x H) (Eq. 1 and 2) or as a function of DBH only (Eq. 3 and 4) for trees in the forest interior and forest edges. Effects of fine-scale architectural variation on allometric relationships (Eq. 1 and 2) were pronounced within the first 76 m from the forest edges, whereas edge effects on Eq. 3 and 4 with woody volume as a function of DBH only were most pronounced within the first 55 m from the forest fragment margins (Supplementary Fig. 8a, b). While equations 1 and 2 reflect differences in allocation patterns at the trunk and branch levels caused by fine-scale architectural variation (i.e. thinning or thickening of branches and trunks, branch loss and changes in branch length), equations 3 and 4 also capture edge effects on tree height (i.e. height growth or height reduction from collateral damages to living trees).

We observed that for a given DBH and height, the surviving tall trees in the edges had similar aboveground woody volume as their counterparts in the forest interior, while short colonising trees in the edges had larger woody volume than those in the forest interior (Fig. 3a). These allometric relationships were also valid for the compartmentalised volume in trunks and branches (Supplementary Fig. 9a, b). However, predictions of woody volume as a function of DBH only revealed a reduced aboveground woody volume of the surviving trees in the edges (Fig. 3b). We found that a considerable fraction of trees near the edges had a disproportionately lower height for a given DBH

(Supplementary Fig. 10), which may have negatively affected the woody volume predictions when using DBH only.

To illustrate these allometric effects, a surviving tree with 33 m height and 70 cm DBH is predicted to have a woody volume of 7.7 m³ in interior forests and 7.4 m³ in the forest edges, a non-significant variation in the woody volume of surviving trees related to edge effects on architectural traits (CI$_{95\%}$: 7.3–8.0 in the interior vs. 7.0–8.0 in the edges). A colonising tree with 10 m height and 10 cm DBH is predicted to have a woody volume of 0.12 m³ in the interior and 0.18 m³ in edges, an increase of 50% in the woody volume caused by edge effects. However, when woody volume is predicted as a function of DBH only, a 70 cm DBH tree in the interior is predicted to have a woody volume of 8.14 m³, but edge effects reduce a surviving 70 cm DBH tree to a volume of 6.27 m³, which corresponds to a 30% decline in its woody volume. This also demonstrates the ability of the allometric model to capture variation in tree height caused by edge effects.

$$\ln(\text{Woody volume}_{\text{interior}}) = -0.21 + 0.81 \ln(\text{DBH}^2\text{H}_{\text{interior}}) + \varepsilon_i \quad (1)$$

$$\ln(\text{Woody volume}_{\text{edge}}) = -0.13 + 0.72 \ln(\text{DBH}^2\text{H}_{\text{edge}}) + 0.016 \ln(\text{DBH}^2\text{H}_{\text{edge}})^2 + \varepsilon_i \quad (2)$$

$$\ln(\text{Woody volume}_{\text{interior}}) = 2.80 + 1.97 \ln(\text{DBH}_{\text{interior}}) + \varepsilon_i \quad (3)$$

$$\ln(\text{Woody volume}_{\text{edge}}) = 2.45 + 1.72 \ln(\text{DBH}_{\text{edge}}) + \varepsilon_i \quad (4)$$

### Edge effects on AGB estimates across larger-spatial scales

We used allometric models to predict woody volume of trees in edge and interior forests, enabling us to estimate aboveground biomass (AGB) across larger spatial scales. Linear mixed models applied to data from 44 1-ha permanent plots revealed a statistically significant reduction in AGB of 24.7 Mg ha⁻¹ due to edge effects ($t = -3.1$; $P$-value = 0.003). This reduction accounted for nearly 10% of the AGB of structurally intact forests (282.2 ± 15.3 Mg ha⁻¹) and comprised two components: first, there was an 18.7 Mg ha⁻¹ decline in AGB due to edge

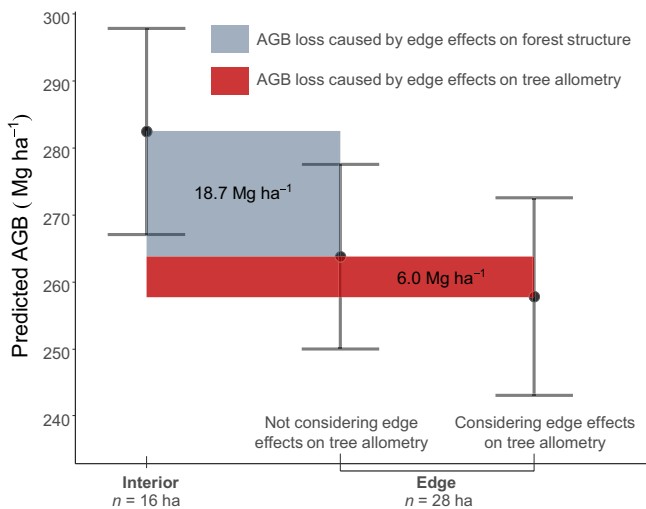

**Fig. 4 | Predicted aboveground biomass (AGB) for edge (N = 28 ha) versus interior (N = 16 ha) forest plots within the Biological Dynamics of Forest Fragments Project (BDFFP) in Central Amazonia, the world's longest-running experimental study of habitat fragmentation.** Data are presented as mean predicted AGB values (black points) using linear mixed modelling and error bars denote 95% confidence intervals. 44 1-ha plots used for AGB predictions contained tree measurements on more than 12,000 individual stems ≥10 cm diameter across 1026 tree species. The shaded dark grey area corresponds to AGB loss after 40 years of fragmentation caused by changes in forest structure owing to edge effects on tree mortality, growth and recruitment. The red area corresponds to AGB loss caused by edge effects on tree allometry, calculated by comparing AGB estimates using an allometric model that considered edge effects on tree allometry with an allometric model developed for interior forests.

effects on forest structure—caused by fragmentation-related variation in tree size, tree density and species composition—within 100 m from the forest margins; second, there was a 6.0 Mg ha⁻¹ decline in AGB caused by edge effects on tree allometry within 55 m from the forest margins. These distance thresholds were chosen based on previous studies indicating stronger edge effects within 100 m from forest edges on forest structure, and our own study indicating stronger edge effects on tree allometry within 55 m from the forest edges (see Methods for detailed explanations). We visually represented the predicted reduction in AGB caused by edge effects on both forest structure and tree allometry, comparing them to control plots in interior forests. Notably, tree allometry alone contributed to one-third of the total AGB decline resulting from edge effects (Fig. 4).

## Discussion

High density terrestrial LiDAR combined with long-term tree measurements across Central Amazonia provided a fresh perspective on the architecture and allometry of trees and their associated impacts on the aboveground biomass of forest fragments. We found that edge effects on tree architecture impacted allometric relationships for predicting woody volume as a function of stem size, and despite the large biomass allocation in the aboveground compartments of young trees, the lower height of tall surviving trees near the edges led to a decline in the aboveground biomass of forest fragments. We estimated an overall 10.0% reduction in AGB (24.7 Mg ha⁻¹) owing to edge effects on the forest structure (caused by mortality, growth and recruitment of trees, 18.7 Mg ha⁻¹) and tree allometry (6.0 Mg ha⁻¹). These numbers indicate that altered tree allometry alone accounted for one-third of all biomass losses caused by edge effects on these forests in Central Amazonia. These findings demonstrate the value of terrestrial LiDAR surveys to challenge some of the assumptions about how much carbon trees store, and to allow the detection of fine-scale changes in tree

architecture. 3D measurements of tree size and shape provide a perspective on forest structure and its spatial variability that is difficult to achieve with destructive methods of tree measurement[2,36].

Forest fragmentation had strong edge effects on architectural traits relative to trees in structurally intact forests. However, these effects were primarily dependent on which stage of their lives trees started experiencing edge effects. Tall trees above 20 m in height were likely established before forest fragmentation in 1980. These surviving trees were able to allocate a similar amount of wood to branches and trunks in comparison to trees in interior forests (Supplementary Figure 9). The forest edges of our study have temperatures 5 °C hotter and lower water availability than the interior, which leads to higher branch loss near the edges[28]. However, we found that surviving trees have the ability to maintain their woody volume, possibly by producing new branches (resprouting), which has positive influences on tree growth and survival[38]. The tall trees in the edges had lower path fraction and were more symmetrical than trees in interior forests. In fact, trees under stress can lose terminal branches[39], with the remaining branches shorter in path length to increase hydraulic safety by decreasing the length that water and nutrients need to travel for photosynthesis[7,40]. The observed higher symmetry of surviving trees near the edges can increase their mechanical stability to high air turbulence caused by strong winds in the forest edges[29,30,41,42]. These results suggest that surviving trees had mechanisms of acclimation to stressful conditions near the forest edges - in particular higher temperatures, lower water availability, stronger winds, with minimal effects on allocation of biomass to the aboveground compartments. However, a complementary explanation could be that edge effects selected for individuals or species that were more adapted to the edge conditions. Indeed, forest fragmentation has led to a high mortality rate of tall trees[23], and the surviving trees may be those with traits that provide better fitness to the micro-environmental conditions near the forest edges. Better understanding the role of species and their intra-specific controls would help us predict variation in tree architecture with forest fragmentation in the Amazon.

The new recruits that colonised the edges of the forest fragments were adapted to maximise light capture given their higher woody volume in the branches and trunks, higher asymmetry and higher path fraction. The forest edges had multi-stemmed trees with deeper crowns because of the high light availability near the edges[17]. Short trees near the edges were more asymmetrical, which may help them capture more light by shifting their biomass towards the forest edges[43]. However, this may lead to a lower mechanical stability[42], and trees must increase the volume of trunks prior to growing asymmetrical crowns to avoid damage[44], a strong predictor of tree mortality in fragmented Amazonian forests[19]. These colonising trees also produced numerous branches of high path lengths, which may lead to lower hydraulic efficiency, as these trees have longer paths to transport water and nutrients for photosynthesis[7]. These architectural changes, alongside a large production of new leaves throughout the year observed in the same forest edges[28], may explain the increased growth rates of understory tree species in the forest edges of the BDFFP project in comparison to the same tree species growing in the understory of interior forests[45], despite the hot temperatures and lower water availability in the forest edges[28]. These results illustrate the ability of colonising trees in forest fragments to capture light and grow, which demonstrates the importance of protecting forest fragments for carbon cycling. However, will these colonising trees be able to grow tall and have high rates of survival in the future? These traits that maximise construction costs (i.e. high asymmetry and high path fraction) in the light-rich environments near the edges may also lead to higher risks of mechanical damage and hydraulic failure. Additionally, as trees grow taller, increasing leaf water stress due to gravity and path length resistance can limit leaf expansion and photosynthesis, and consequently limit further height growth[46]. Long-term monitoring of

tree architecture in controlled fragmentation experiments, including the BDFFP, is crucial to predict how these understory trees may survive in fragmented forests when they grow older and, perhaps, taller.

Allometric models used to predict woody volume based on stem diameter and tree height were affected by edge effects. Short trees colonising the edges exhibited woody volume up to 50% higher than trees with similar diameter at breast height (DBH) and height in the forest interior. However, although surviving tall trees for a given height and DBH were able to maintain their woody volume under edge effects, some tall trees had a disproportionately lower height for a given DBH (Supplementary Figure 10). This resulted in a 30% decline in the woody volume of large trees, possibly caused by. the higher microclimatic stress, increased wind speeds, proliferating lianas, and the mortality of large trees and branch loss[23,28,47] that may have led to collateral damage to neighbouring trees, resulting in reduced heights near the forest edges[31,48,49]. Alternatively, trees in the edges may have grown less in height, or edge effects may have favoured shorter individuals or species acclimated to edge conditions, as the height-to-diameter ratio is a strong determinant of mechanical safety[50]. Consequently, a potential combination of stem breakage from damages caused by neighbouring trees and selective forces favouring shorter trees for a given DBH led to reduced woody volume in large trees. This aligns with the significant negative effects of human-induced disturbances on tree biomass[31].

Variation in the woody volume of trees, coupled with variation in the wood density caused by shifts in species composition, determine the spatial patterns of AGB, which vary considerably across undisturbed Amazonian forests. Intact forests at a site about 15 km south of our study area, in similar lowland forests, were found to have mean AGB values of $325.5 \pm CI_{95\%} 13.6$ Mg ha$^{-1}$ in comparison to our estimates of $282.2 \pm 15.3$ Mg ha$^{-1}$[51]. This demonstrates that our estimates of AGB are conservative given their lower values relative to other estimates for the same region. Indeed, AGB estimates can vary significantly among equations, as the choice of allometric models and measurement uncertainty can lead to uncertainties of up to 31% and 16%, respectively, in the AGB estimation of Amazonian forests[52]. Furthermore, local environmental conditions that vary at small spatial scales, such as soil fertility, can account for a third of the variation in AGB in terra-firme Amazonian forests[53].

Forests in edges tend to experience a dramatic decrease in AGB caused by changes in forest structure, and are often structurally similar to secondary forests[54]. We observed a reduction of 18.7 Mg ha$^{-1}$ (6.6%) in the AGB of forest fragments owing to changes in forest structure within the first 100 m from the forest edges. This reduction resulted from combined edge effects on tree mortality, growth and recruitment that influence tree size, tree density and species composition. Forest fragmentation increases the mortality of large trees[22,55,56], but can also lead to higher growth rates of large trees and the recruitment of new individuals[57]. However, large tree mortality represents a significant proportion of biomass loss, contributing to the high carbon emissions of fragmented ecosystems[22,58,59]. The observed reduction in AGB in our study area near the edges was lower than anticipated, potentially due to the controlled conditions in the BDFFP that minimise additional anthropogenic influences such as illegal logging, hunting, fire penetration and pollution[60]. Indeed, edge effects on forest structure in Amazonian forests can extend up to 1000 m, as fragments are more susceptible to recurring disturbances from fires and illegal logging, and sensitive to the land use in the matrix surrounding the forest fragment[61]. Moreover, edge effects can be stronger in the initial decades post-fragmentation, with up to 36% of the forest biomass lost in the first 10 to 17 years after fragmentation, potentially followed by a recovery in the subsequent decades[22,62]. Long-term monitoring of edge and interior forest plots is crucial to investigate edge effects on AGB dynamics, as confounding environmental factors, such as soil and topography, can shape the species composition and forest structure, and thus influence the recovery of fragmented forests.

Edge effects on tree allometry further intensified AGB loss, contributing to an additional 6.0 Mg ha$^{-1}$ AGB reduction in Central Amazonian forests. This accounted for one-third of all AGB losses (24.7 Mg ha$^{-1}$) in these forests 40 years of forest fragmentation—a value previously unquantified in the literature. These values, however, could be conservative, considering that edge effects on tree allometry may penetrate farther into the forest interior ( > 55 m) across other sites in Amazonia and lead to more significant negative impacts on forest AGB[60]. Given that fragment edges cover a total area of 176,555 km$^2$ of Amazonian forests[58], our findings of edge effects on tree architecture and allometry could translate into a substantial component of fragmentation-related carbon losses.

The decline in AGB near the edges presents an apparent contradiction with the higher Plant Area Index (PAI) found in these forest edges[24]. Specifically, Maeda and colleagues (2022) found higher PAI values, a combination of leaf and wood surface areas, near the forest edges due to a large increase in the density of small trees. This increase compensated for the reduction of upper canopy PAI caused by the mortality and damage of tall trees. This contradiction can be explained by the disproportionate contribution of tall trees to the aboveground forest biomass[63,64]. Furthermore, the edges of these fragments are dominated by pioneer species with acquisitive traits[65], thriving under the light-rich environment owing to lateral light penetration and gap formation associated with the mortality of large trees[66]. Traits of pioneer species include a larger volume per mass unit (low wood density) and large foliar area[67], which together increase the PAI of edge forests without a concomitant increase in AGB[68]. This highlights the challenge faced by passive sensors onboard satellite platforms to capture a potential reduction in AGB in fragmented forests with increasing PAI.

Despite our efforts to understand architectural variation in Central Amazonian forests with edge effects, the mechanisms allowing species to change their architecture remain elusive. The within-plot variability, that can be seen as local effects arising from edge effects, as well as species-specific and ontogenetic influences, accounted for most of the architectural variability. These results may help elucidate the generality of our findings in architectural variation across Central Amazonia, although large-scale Amazonian gradients in topography, edaphic properties and climate remain to be tested. Moreover, a shift in species composition caused by edge effects may have partially contributed to our observed changes in tree architecture[65]. We propose that future research on tree architecture should continue to unravel the interactions of the environment with functional diversity within species. TLS-based data, combined with molecular, genetic and physiological processes regulating tree architecture, can help resolve debates concerning the mechanisms by which trees change their architecture[1,69], which may enhance predictions of plant responses to global changes. Understanding these mechanisms is crucial for predicting the ability of plants to survive and grow with increasing forest fragmentation, and their contribution to the terrestrial carbon sink dynamics.

## Methods

### Study site and sampling design

The study was conducted in Central Amazonian forests (2°20 30 ′S, 60° 05 37 W) near Manaus, Brazil, in the reserves of the Biological Dynamics of Forest Fragments Project (BDFFP), the world's longest-running experimental study of habitat fragmentation[37]. The region has seen notable carbon and biodiversity losses due to forest fragmentation effects[58,65]. The BDFFP sites consist of forest fragments that were initially isolated in 1980-1983 following the conversion of the surrounding mature forest into cattle pastures. The project was specifically designed to investigate the relationship between the size of a forest fragment and its stability and functioning. The experimental

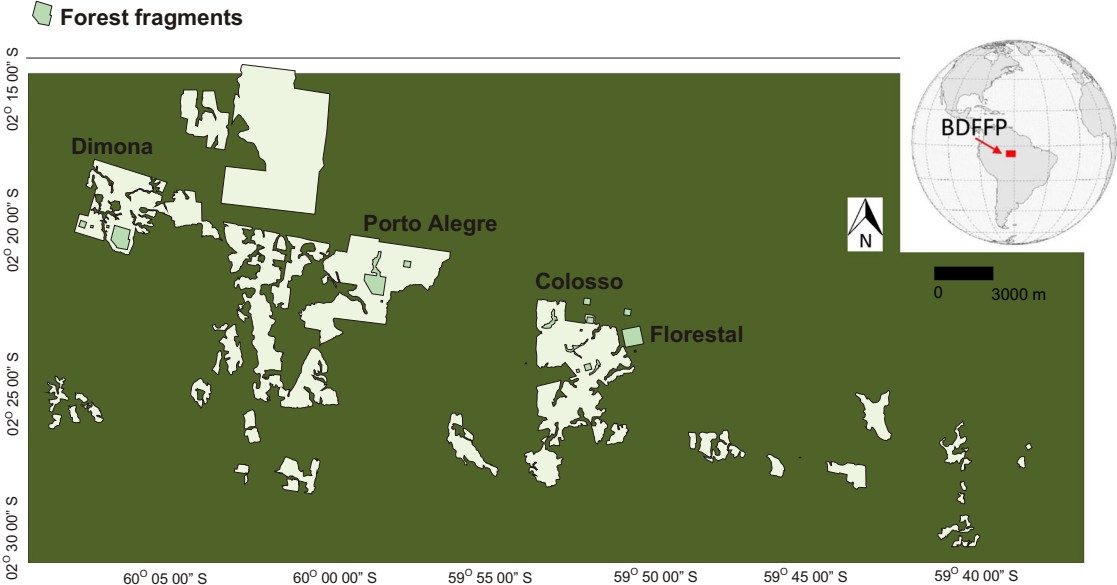

**Fig. 5 | The Biological Dynamics of Forest Fragments Project (BDFFP), the world's longest-running experimental study of habitat fragmentation, is located in Central Amazonia.** The BDFFP sites are composed of forest fragments originally isolated in 1980−1983 after the conversion of the surrounding mature forest into cattle pastures. The forest fragments are surrounded by a 100 m matrix, regularly cleaned by cutting the regrowth vegetation to keep the forest fragment isolated.

design of BDFFP is based on comparisons of a replicated series of forest fragments or reserves of different sizes before and after their isolation from continuous forests[70]. Currently, the matrix surrounding the forest fragments is dominated by secondary growth forests, but a 100 m strip surrounding the forest fragments is cleared regularly by cutting vegetation regrowth to maintain their isolation (Fig. 5). As an experiment that minimises additional anthropogenic influences such as illegal logging, hunting, fire penetration and pollution, the project offers insights into ecological and environmental changes in fragmented forests.

For this study, we investigate how the architecture of trees varies with distance from the forest edge in four forest fragments across two regions in Central Amazonia (Dimona and Colosso; two of 1 ha, one of 10 ha, and one of 100 ha reserves). The maximum distance from forest edges towards the forest interior varies between fragments, with 50 m for 1 ha fragments, 100 m for the 10-ha fragment and 500 m for 100 ha the fragment. Compared to undisturbed forests in the fragment interior, the forest near the edges is dominated by a higher density of early successional, fast-growing species because of elevated tree mortality and seed dispersal from degraded neighbouring habitats[37,65]. The edges of these fragments can have up to 5 °C higher temperatures than the forest interior during the dry season[28]. During periods of lower precipitation in the region, the edges of these fragments have lower soil moisture, which creates periods of higher evaporative demand for tall trees in the edges[28]. The fragment edges have a sparser upper canopy marked by lower plant area index (PAI) above 15 m in vertical height, which creates a light-rich environment and higher light availability in the understory of forest edges[24].

**Terrestrial laser scanning: data acquisition and pre-processing**
The three-dimensional structure of trees was assessed based on point-cloud data acquired using a state-of-the-art terrestrial laser scanning (TLS) system. The TLS data were acquired using a RIEGL VZ-400i system in April 2019 within the BDFFP permanent forest plots. We used a scan resolution of 40 mdeg in both azimuth and zenith directions, which results in a point spacing of 34 mm at 50 m distance from the scanner. The laser pulse repetition rate used was 600 kHz, allowing a measurement range of up to 350 m and up to eight returns per pulse.

The scans covered three transects of 100 × 10 m and three transects of 50 × 10 m near the fragment edges and perpendicular to the forest fragment margins (Fig. S1). We also included one transect of 30 × 10 m length in the centre of the 100-ha forest fragment, which lies 500 m from any fragment margin to ensure sampling of forest interior in the absence of edge effects on the tree architecture.

To ensure a full 3D representation of the upper canopy (maximum canopy height = 36 m), each transect consisted of three scan lines parallel to each other with scans spaced at 5 m intervals within and between lines (Supplementary Fig. 1). The distance between scanning positions was smaller than the 10–40 m usually applied in previous studies to minimise data uncertainties due to occlusion in dense tropical forests and maximise data acquisition in the upper canopy[70]. Given that the RIEGL VZ-400i has a zenith angle range of 30–130°, an additional scan was acquired at each sampling location with the scanner tilted at 90° from the vertical position. A total of 1,188 scans across all transects resulted in a complete sampling of the full hemisphere in each scan location (Supplementary Figure 1). All scans were later co-registered into a single point cloud per transect using the RiSCAN PRO software version 2.9. Given that the RIEGL VZ-400i uses onboard sensor data with an algorithm to align scans without the use of reflectors, automatic registration was done before a final adjustment of scans. More details on how the data was used to investigate edge effects on structure and dynamics can be found in Maeda and colleagues[24] and Nunes and colleagues[28], respectively.

**Individual tree segmentation and quantitative structural modelling**
Individual tree segmentation used an automatic approach followed by manual corrections. The automatic segmentation of individual trees was based on the shortest paths method by Raumonen and colleagues[71]. In this method, the shortest paths from the points to the base layer above the ground level are determined with restrictions on horizontal directions allowing better dealing with occluded regions and holding back paths going sideways into neighbouring trees. After the shortest paths were determined, the trees are defined as all points connected via shortest paths to the same stem section close to ground. The stem sections are iteratively defined around the base layer points that are

**Table 1 | Traits, structural characteristics of trees with high trait values and their ecological meaning. This table summarises the information presented in more detail in the section "Tree architectural traits" of the Supplementary Methods 2**

| Trait | Structural characteristics | Ecological interpretation |
|---|---|---|
| Branch/trunk surface area divided by branch/trunk woody volume | Trunk and branch thickness | High trait values indicate a higher proportion of metabolically inactive wood[6]. This is linked to lower support of aerial structures for light capture and productivity. On the other hand, higher values indicate higher interaction with the atmosphere, including higher respiration rates of the living tissues[5]. |
| Relative crown width | Horizontal crown size relative to DBH | High trait values indicate trees that develop wide crowns, usually linked to limited access to light[77,86], although it can be limited by competition from neighbours[87]. Shade-tolerant trees can also expand their crowns to maintain minimal leaf overlap for light capture and withstand falling debris[88]. |
| Relative crown depth | Vertical crown size relative to tree height, often caused by multi-stemming | In high-light environments trees invest in deep crowns to better compete with neighbours[17]. Multi-stemming also leads to large relative crown depths, as our algorithm also considers the crown as of the upper segments from any multi-stemming point. |
| Path fraction | Umbrella-shaped tree crown | Longer paths of trunks, branches and twigs prioritise sun exposure and light capture. This leads to longer water and nutrient transport distances that require high costs related to construction tissue[7]. High path fraction values represent umbrella-shaped crowns that prioritise sun exposure and light capture but are structurally expensive to build and are hydraulically less efficient[10]. |
| Asymmetry | Wood allocation in branches and trunks shifted relative to the trunk's vertical axis | Tree asymmetry is a result of competitive pressure from neighbouring trees[78]. Asymmetrical crowns can maximise capture of solar radiation by shifting their trunks and branches towards canopy gaps or away from their neighbours to avoid competition[43]. However, asymmetrical trees have lower mechanical stability and are more vulnerable to winds[30], which is a major cause of mortality of fragmented Amazonian forests[19]. |

ranked by the number of shortest paths connected to the point. It started from the point with most paths, the stem section around was first defined, and proceeded with the next highest ranked point not yet assigned to a tree. Considering a reverse J-shaped distribution of tree size of Amazonian forests - with a decreasing density of individuals with increasing diameter at breast height (DBH)[72], we started the extraction of trees from the largest to smaller individuals within each transect to ensure that most large trees within the study area were included. However, some limits were imposed to the automatic approach, including (1) the dense canopy with trees occupying different strata, (2) the high diversity and variability in tree architecture, (3) the high abundance of lianas and (4) the noise in the data produced by wind flapping. Manual corrections to remove lianas and to identify extraneous and missing branches of trees represented a substantial fraction of the processing time. Trees that were severely damaged or dead, as well as trees whose crowns could not be clearly discerned from those of lianas or neighbouring trees were excluded from our analysis.

A total of 315 individual trees were segmented and extracted from the original TLS point clouds. A quantitative structure model (QSM) of the woody structure, proposed by Raumonen and colleagues[73], quantitatively describes the topological, geometric and volumetric properties of trees (Supplementary Fig. 1). A QSM consists of a hierarchical collection of building blocks—usually geometric primitives such as cylinders and cones—which are fitted to the point clouds to locally approximate the woody parts. The use of circular cylinders in our study is a robust and accurate approach to estimate diameters, lengths, directions, angles and volumes[74]. Fitting cylinders to smaller branches, however, can lead to significant overestimation of the diameters and volumes stemming from the uncertainties of LiDAR measurements[75]. To reduce these uncertainties related to branch size, we followed Jackson and colleagues[42] and trimmed off branches with diameters <2 cm from the QSMs. This procedure was reported to not affect the estimate of architectural traits across tropical forests[42]. QSMs provided estimates of branch size distribution, parent-child relations of the branches, diameters, lengths, angles, directions, and volumes, which are key properties that can be used to estimate architectural traits[10]. Details of parameters for QSM reconstruction can be found in Supplementary Methods 1. The QSM database included

trees that varied in DBH (2.4–72.3 cm), tree height (5.0–35.8 m) and volume (0.01–13.0 m$^3$) (Supplementary Fig. 2).

## Tree architectural traits

In this study, we propose that surface area per unit volume (m$^2$ m$^{-3}$) of branches and trunk, asymmetry [-], path fraction [-] and relative crown dimensions, including relative crown depth (m m$^{-1}$) and relative crown width (m cm$^{-1}$), are traits that may be sensitive to forest fragmentation, as they have been linked to maintenance costs for respiration, mechanical stability, hydraulic conductance and light capture. Retrieval of architectural traits are described in detail in Supplementary Methods 2. The ecological meaning of each architectural trait is synthesised in Table 1, and in more detail in Supplementary Methods 2.

## Woody volume and allometric relationships

The accuracy of allometric models to predict aboveground biomass is crucial for terrestrial carbon stock mapping. Total tree height co-varies with bioclimatic stress that depends on temperature and precipitation variability[76,77], but little is known how forest fragmentation may affect these relationships. Locally derived DBH-height relationships improve existing allometric equations[77]. Differences in wood allocation in branches caused by fragmentation-related environmental changes may lead to differences in equations that predict woody volume as a function of DBH and total tree height[78]. In this paper, we investigated the effects of fragmentation on allometric relationships between DBH$^2$H and woody volume, and also on allometric relationships between DBH and total woody volume, and discussed the implications of potential effects on the mapping of terrestrial carbon in fragmented forests. Woody volume, DBH and tree height were derived from the tree's QSMs. The QSM method can provide accurate woody volume estimates with no systematic bias regardless of the tree structural characteristics across tropical forests in Cameroon, Peru, Indonesia, and Guyana[79,80]. Therefore, nondestructive estimates of woody volume from TLS can be a replacement for traditional sampling methods, and for updating allometry and reducing uncertainties in landscape-level biomass estimates[81].

## Tree height to classify trees by time of establishment before or after forest fragmentation

Forest fragmentation may affect each architectural trait differently depending on the ontogenetic stage and tree size. As the fragment isolation occurred in 1980, the tall trees of our dataset are individuals that likely have survived the fragmentation effects, and differences in their architecture with forest fragmentation may reflect the ability of adult plants to acclimate to edge effects. On the other hand, short trees can be included in three groups: (1) ~2/3 of these trees are less than 30 years old recruits that have colonised the area after the establishment of the forest fragments; (2) surviving trees from understory species at varying ontogenetic stages; and (3) trees from slow-growing species, including upper canopy species at varying ontogenetic stages. Trees in group 1 have already been exposed to edge effects during their juvenile phase. Conversely, changes in the architecture of trees in groups 2 and 3 reflect acclimation of short-stature adult individuals to edge effects that, analogous to the tall surviving trees, change their architecture during the adult phase.

Long-term measurements from the BDFFP project since 1980 revealed that, among all the trees that have been tagged and monitored, all trees > 20 m in height have been growing in the BFFDP forest fragments before 1990 (Supplementary Methods 3; Supplementary Fig. 3), which indicates that trees > 20 m in height are those that have survived the fragmentation effects. Therefore, any change in the architecture of these tall trees must have occurred during their adult phase. Our data also indicated that ~ 33% of trees <20 m in height have also been growing in these forests since before 1990. This suggests that 66% of the small trees <20 m in height are new, young recruits (i.e., trees that have colonised these fragments within the last 30 years) that were exposed to edge effects already in their juvenile phase. We thus used 20 m tree height as a threshold separating colonising short trees ( < 20 m) and surviving adult trees ( > 20 m).

## Determining edge effects extent on architectural traits

Long-term studies from the BDFFP have demonstrated that edge effects vary widely in extent, with biophysical changes that can penetrate from 20 m to 300 m[37]. To estimate the extent of edge effects on each architectural trait, we used mixed linear models (LME, Eq. 1) that contained a variable representing the plot category of location near an edge or in the forest fragment interior (edge effects), following Nunes and colleagues[82]. The model also included a variable that represented tree height (H), considering that architectural traits co-vary with tree size and ontogenetic stage[10,83]. Edge effects and tree height were treated as additive terms to examine the significance of fragmentation and tree height on the variation of architectural traits. We also included an interaction term edge effects × tree height, as fragmentation may have different effects on trees of different heights (Eq. 5). To examine the influences of distance from edges on the allometric relationships between woody volume and DBH$^2$ H or woody volume and DBH, we tested how the log transformed variables interacted with edge effects (Eq. 6 and Eq. 7). Nested effects of forest site (Colosso versus Dimona sites), fragment size (1, 10 or 100 ha) and plot (or transect) identity were treated as random variables ($\mu$), allowing us to account for the nested spatial variation in architectural traits and to include any idiosyncratic differences between forest site, fragment size and micro-environmental variation (i.e., topography, soil) between plots.

$$\text{Architectural trait} = \beta_0 + \beta_1 \times (\text{edge effects}) + \beta_2 \times H + \beta_3 \times (\text{edge effects}) \times H + \mu_i + \varepsilon_i \quad (5)$$

$$\ln(\text{woody volume}) = \beta_0 + \beta_1 \times (\text{edge effects}) \times \ln(\text{DBH}^2 H) + \mu_i + \varepsilon_i \quad (6)$$

$$\ln(\text{woody volume}) = \beta_0 + \beta_1 \times (\text{edge effects}) \times \ln(\text{DBH}) + \mu_i + \varepsilon_i \quad (7)$$

where $\beta_0$ to $\beta_3$ are the model parameters, $\mu_i$ is the random intercept for the nested effects of region i, fragment size i and plot identity i, and $\varepsilon_i$ is the normally distributed residual error.

We then tested the influence of distance to edges on each architectural trait with distances to edges varying between 1 and 100 m, including tree woody volume. We then tested the influence of distance to edges, varying from 1 to 100 m, on each architectural trait; we determined the edge effects extent for each architectural trait based on the maximum absolute t-value of the term "edge effects" of the model (Supplementary Method 4; Supplementary Fig. 4). The edge effects extent was then used to categorise our analysis into edge versus interior trees during all trait analyses.

We also examined the explained variance by the random variables of Eq.5 to investigate the spatial variability of architectural traits arising from region, landscape and plot (Supplementary Fig. 5). The LME models were fitted using the lme function in the "nlme" R package.

## Statistical modelling to predict edge effects on architectural traits

We used nonlinear mixed-effects models of architectural traits of trees in forest edges and trees in forest interior as a function of tree height (Equation 8). Allometric relationships between woody volume and DBH$^2$ H or between woody volume and DBH were fitted using linear mixed models after log-transforming the dependent and independent variables[84] (Equations 9 and 10). We tested whether a quadratic term should be included to increase goodness-of-fitness of the allometric models by comparing each model's AIC (Supplementary Table 1). For all the models, plot identity nested within landscape (position of fragment within the landscape and fragment size) and region within Central Amazonia were included as random variables, allowing us to include any idiosyncratic differences between plots, fragments and regions. The models were fitted using the nlme function for the nonlinear mixed-effects model and *lme* function for the linear mixed-effects model in the "nlme" R package. Performance of the final models was also evaluated using an 80/20 split of the data for calibration and validation, respectively, over 200 randomised permutations of the dataset. These analyses generated a distribution of model coefficients and allowed an assessment of model stability and uncertainty of predictions.

$$\text{Architectural trait} = \beta_0 \text{Tree Height}^{\beta 1} + \mu_i + \varepsilon_i \quad (8)$$

$$\ln(\text{Woody volume}) = \beta_0 + \beta_1 \ln(\text{DBH}^2 H) + \beta_2 \ln(\text{DBH}^2 H)^2 + \mu_i + \varepsilon_i \quad (9)$$

$$\ln(\text{Woody volume}) = \beta_0 + \beta \ln(\text{DBH}) + \beta_2 \ln(\text{DBH})^2 + \mu_i + \varepsilon_i \quad (10)$$

where $\beta_0$ to $\beta_2$ are the model parameters, $\mu_i$ is the random intercept for the nested effects of region i, fragment size i and plot identity i, and $\varepsilon_i$ is the normally distributed residual error. Let DBH in cm, H in m and woody volume in m$^3$.

## AGB calculation

Permanent plots within the BDFFP were distributed across a large area of 1000 km$^2$ in Central Amazonia (Fig. 6). We used 44 permanent plots of 1 ha each in edge (28 ha) and interior forests (16 ha) to compile aboveground biomass (AGB) using data from Dimona, Colosso, Florestal and Porto Alegre. The project was designed specifically to investigate the relationship between the size of a forest fragment and its stability and functioning. The experimental design of BDFFP is based on comparisons of a replicated series of forest fragments or reserves of different sizes before and after they were isolated from continuous forests. We then examined how edge effects on tree allometry control AGB relative to edge effects on forest structure (for

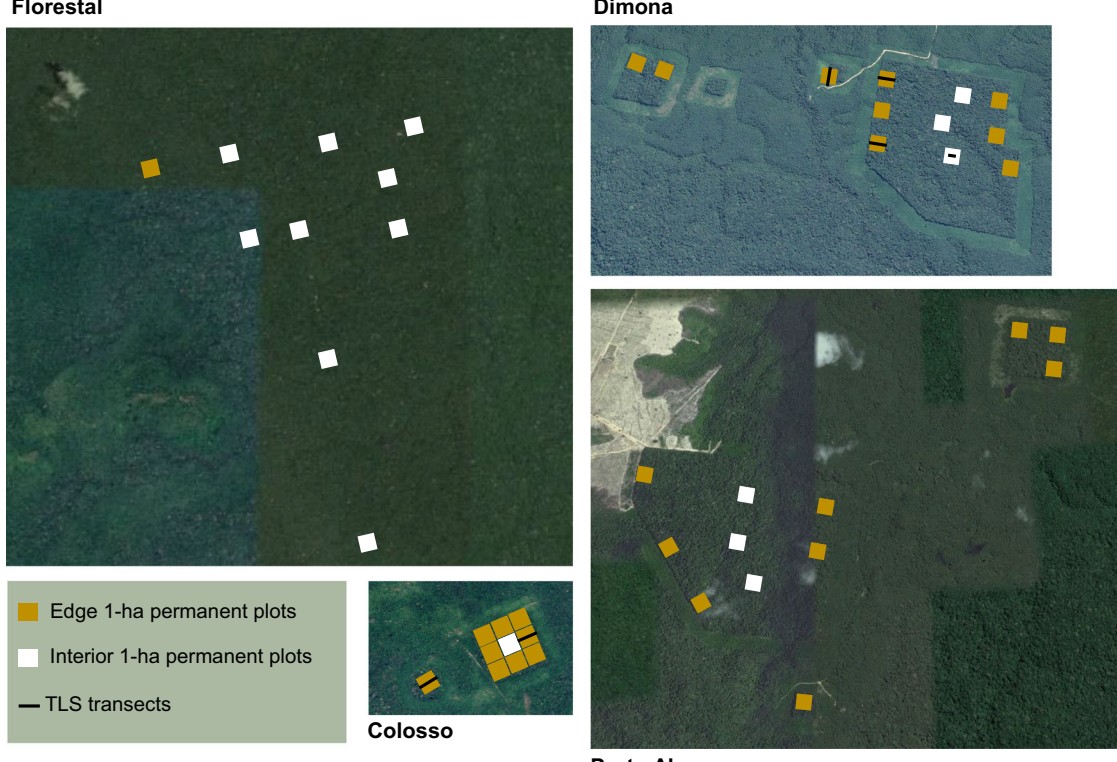

**Fig. 6 | Allometric models for woody volume estimates were developed from tree measurements using a Terrestrial Laser Scanner (TLS) within the Biological Dynamics of Forest Fragments Project (BDFFP) in Central Amazonia.** Six transects (five of 100 × 10 m at the edges and one of 30 × 10 m in the forest interior), denoted as black lines in the Colosso and Dimona sites, allowed us obtain high quality point clouds to segment and extract 315 trees. Tree measurements of stem size combined with species identification across 44 1-ha permanent plots within Florestal, Dimona, Colosso and Porto Alegre were used to estimate the above-ground biomass (AGB) in edge (yellow plots) and interior (white plots) forest plots. Our findings demonstrate that edge effects on tree allometry penetrated 55 m from the forest edges. Comparisons of AGB values using edge- versus interior-specific allometric equations were made to predict the influences of tree allometry on the forest AGB.

example, edges effects on tree density and distribution of tree size caused by variations in tree mortality, growth, and recruitment).

We used diameter at breast height (DBH), defined as 1.3 m from the base of the stem, from 12,112 tree stems. For buttressed stems or other deformities, the point of measurement is raised above the deformity. All trees were distributed in 66 families, 254 genera, and 1026 species. Plants were identified at the species level, or at least at the genus level. We then converted diameter measurements to woody volume estimates using the allometric equations 3 (interior forests) and 4 (edge forests), which predict woody volume as a function of DBH only. These models capture variation in woody volume related to tree height (i.e. tree damage and breakage) and fine-scale architectural variation (i.e. branch production, branch loss, branch and trunk thickening etc). To convert woody volume values to tree-level AGB, we used wood density values extracted from a global wood density database (https://datadryad.org/stash/dataset/doi:10.5061/dryad.234)[85], using the BIOMASS package in R. In cases where a stem was unidentified or where no taxon-specific wood density data were available, we applied the appropriate genus or family-specific wood density values. Aboveground biomass for each 1-ha plot was estimated as the sum of the tree-level AGB.

We employed linear mixed modeling to analyze AGB values, treating a categorical variable as a fixed effect indicating whether plots were near an edge (<100 m from the forest margins) or in the interior (>100 m from the forest margins). This approach allowed us to estimate AGB variation resulting from edge effects. We chose a 100 m threshold to differentiate between edge and interior forest plots due to the higher mortality and turnover rates observed within the first

100 m from the forest edges within the BDFFP[22,55]. Forest site (Colosso, Dimona, Florestal e Porto Alegre) was considered a random variable, accommodating any unique differences between forest sites that could impact species composition and forest structure, such as soil and topography. Differences in predicted AGB values between edge and interior forests using allometric equation 3 (for interior forests) represented variations in forest structure, subsequently affecting forest AGB. Comparing predicted AGB values in forest edge and interior forests using their respective allometric models (Equation 3 for interior forests and Equation 4 for edge forests) enabled us to capture AGB variations caused by edge effects on both forest structure and tree allometry. We calculated 95% confidence intervals based on the uncertainty in model parameters. The linear mixed effects (LME) models were fitted using the lme function in the "nlme" R package.

### Reporting summary
Further information on research design is available in the Nature Portfolio Reporting Summary linked to this article.

## Data availability
The architectural traits data generated in this study have been deposited in the national Finnish Fairdata services database under accession code https://etsin.fairdata.fi/dataset/c71785e6-58ab-4316-a0fc-28236c76bbe7. The point clouds of individual trees can be available by contacting the corresponding authors upon reasonable request within 10 days. Measurements of all trees and their corresponding species identification at the plot level are products of the long-term Biological Dynamics of Forest Fragments Project. The

interested party should contact William Laurance (bill.laurance@jcu.edu.au), Ana Andrade (asegalin@gmail.com) and Alberto Vicentini (vicentini.beto@gmail.com) for potential access to these data.

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

## Acknowledgements

This study was funded by the Academy of Finland (decision numbers 318252, 319905, and 345472 to Eduardo Eiji Maeda). This publication is

number XXX of the Technical Series of the Biological Dynamics of Forest Fragment Forest (BDFFP—INPA/STRI). We thank the Biological Dynamics of Forest Fragments Project for the great logistical support in the field and the research assistance. Jérôme Chave acknowledges "Investissement d'Avenir" grants (CEBA, ref. ANR-10-LABX-25-01; TULIP, ref. ANR-10-LABX-0041). Jin Wu was supported by the National Natural Science Foundation of China (#31922090), the Innovation and Technology Fund (funding support to State Key Laboratories in Hong Kong of Agrobiotechnology) of the HKSAR, China, and the Hung Hing Ying Physical Sciences Research Fund. Josep Peñuelas was supported by the TED2021-132627B-I00 grant, funded by MCIN and the European Union NextGeneration EU/PRTR, and the CIVP20A6621 grant funded by the Fundación Ramón Areces. Gabriela Zuquim acknowledges the Academy of Finland (grant number 344733) and Danish Council for Independent Research—Natural Sciences (grant number 9040-00136B).

## Author contributions

M.H.N. and E.E.M. conceived the study. M.H.N. and E.E.M. led TLS data collection in the field. J.L.C.C., W.F.L., A.A., A.V., and S.L. collected tree measurement data and identified tree specimens. M.H.N., P.R., and E.E.M. processed the LiDAR data. M.H.N. performed data analyses and wrote the manuscript. M.H.N., M.C.V, J.L.C.C., W.F.L., P.R., T.J., G.Z., J.W., J.P., J.C. and E.E.M. contributed to the revision of the paper.

## Competing interests

The authors declare no competing interests.
