## [Peer Review File · Nature Communications]

REVIEWER COMMENTS

Reviewer #1 (Remarks to the Author):

The study presents new insights into the effects of forest fragmentation on above ground biomass and individual tree architecture in Amazonian forests. The main finding is that forest fragmentation is not only reducing above-ground biomass through edge effects on forest structure, but also by affecting tree architecture and allometry. The authors use terrestrial lidar data to quantify tree architectures and allometry and permanent plot data from long-term experimental sites in Amazonia to quantify above-ground biomass.

The study is interesting from two different perspective. First, the study addresses a topic that is relevant to understand general patterns of tree architecture in response to changing environmental conditions. Second, the findings have important implications for quantifying biomass and carbon stocks of tropical forests, because the authors show that changes in tree allometry following forest fragmentation may lead to an overestimation of biomass at forest edges, if differences in allometry are not being accounted for in allometric equations. While I find the study very interesting and basically well conducted, there are a few things that remain unclear. Before considering the paper for publication, it would be very helpful if the authors could clarify a few points that I am outlining in the following.

AGB of forest edges and interior:

In figure 4, the authors report that edge effects on tree allometry cause an additional reduction in AGB, which is one of the key findings. I have a few questions and probably some doubts about that finding.

First, I think I do not fully understand the difference between the AGB estimate for forest edges and the forest edges + allometric effects. Is the estimation of AGB for forest edges based on allometric equations that do not take the allometry of trees at the edges into account, while the additional 6.6 Mg ha⁻¹ result from considering differences in allometry?

Second, the sample sizes for edge and interior plots strongly differ (30 vs. 6). I think it is crucial to clarify how the plot locations were selected (e.g. randomly vs purposefully selected as representative of the larger area). Even in primary forests, forest structures may spatially vary strongly (e.g. Ehbrecht et al. 2021). With increasing sample size, usually more variability is being captured during sampling. That makes me wonder whether the differences in AGB could just be an artefact resulting from differences in sample size. How do the AGB estimates of edge and interior compare to Amazon-wide estimates of AGB? I think the discussion could benefit from providing a few more references on how edge effects on AGB compare to findings from other studies.

Third, I do not fully understand how the authors define forest edges. In Supplementary Figure 1, the BDFFP study site is shown with different transect length for edge (100 m) and interior (30 m) forests. Does the length of these transects represent the area of forest edges (in terms of depth/width) used for comparing AGB between interior and edges? In the results section, the authors state that they compared AGB estimates within 55 m from forest margins to predict impacts of edge effects on tree allometry and subsequently AGB. In the next paragraph, they state that within 100 m from forest margins AGB is reduced compared to interior forest. Some of the 100 m transects shown in

Supplementary Figure 1 cover the entire forest patch (where is the interior part then?). First, I am wondering why the authors did not use the information from the entire 100 m transect for the allometry part? Second, it remains unclear whether AGB and interior structure in 1 ha and maybe 10 ha fragments are comparable to those of 100 ha fragments, as the interior parts in the small fragments may also already show differences in allometry? Other studies for example found deep reaching edge effects of up to 200 m (see Nguyen et al., 2023, <https://doi.org/10.1007/s10980-023-01609-x>). Overall, the study design descriptions leave me a bit puzzled. I think the paper could benefit from a more straightforward study site and study design description. For example, I would recommend to include a map that shows the location of the plots that were used to estimate AGB for edges and interior and provide a clear definition of forest edge in terms of area and distance from the forest margin.

Tree size thresholds:

In the results section about edge effects on architectural traits, the authors mention that trees above 20 m in height were considered to have established before forest fragmentation and that trees below 20 m were most likely post-fragmentation recruits. In Supplementary Figure 4, the authors show that is a considerable number of trees smaller than 20 m (in 2019) that were first measured in the early 1980's. Doesn't this imply that the threshold is rather arbitrary and that there is likely a not too small proportion of trees that are < 20 m but that recruited before the experiment was established? Wouldn't it make sense to derive such a threshold from tree growth models? How would different thresholds affect the results?

Competition:

In different sections and paragraphs of the text, the authors state that crown asymmetry indicates the capability to avoid competition for light. I just want to highlight that there are also studies showing that crown asymmetry can also be considered a measure of competition intensity and not necessarily as a measure of competition avoidance (Seidel et al. 2011 <https://doi.org/10.1016/j.foreco.2011.03.008>).

Tree heights:

At several points, the authors speak of reductions in tree height in forest edges. That term always left me a bit confused when reading the manuscript. As the authors mention, trees can be shorter near edges because of damages or mortality of upper crown parts. But is this generally the case for the studied trees in the study sites? Speaking of reduction in tree height in edges implies that something has reduced the height, e.g. a damage. But couldn't it be that some trees are simply shorter due to the different environmental conditions near edges? For example, under better light conditions trees might need to invest less into height growth in order to capture light. In case there is a considerable number of trees that are not shorter because of damages/crown mortality, but rather of different growing conditions or simply age, I suggest to rather speak of "lower" tree heights instead of "reduced" or "reduction".

Minor comments:

The meaning of the last sentence of first paragraph of methods section is unclear.

Summarizing, I think it is a very relevant and interesting study. However, I see some room for improving the manuscript, especially by a clearer description of the study design. Methods sections should be written in a way that a study is reproducible. In the current form, I would find it difficult to reproduce the study due to a lack of clarity. As a reader, I was always jumping back and forth between results, methods, discussion and SI to get how and what the authors did (which left me a bit confused here and there). By streamlining the outline and text the manuscript could be greatly improved I think. I hope my comments and suggestions can help to improve the manuscript.

Reviewer #2 (Remarks to the Author):

Amazonian forests are becoming more fragmented. This manuscript employs TLS across Amazonian sites to assess edge effects on tree architecture and allometry/biomass. In a nutshell, authors found that biomass increases due to enhanced light capture were more than offset by branch loss due to increased susceptibility to disturbance and damage at forest edges (eg higher wind turbulence) for new (colonizing) and established individuals.

TLS-based studies like this are increasing in prominence and offer direct methods to measure individual tree biomass and associated biomass without destructive harvest. Results are novel, and the implications on large-scale biomass estimation potentially transformative. That said, I think that it is important to recognize some of the studies limitations: especially its limited sample size and extent as well as relative inattention to the role of inter-species trait differences. For the most part, species are lumped and I wonder how limiting factor would affect a wider ranging study. These limitations are part of the nature of doing labor-intensive, fine-scaled TLS scans so that's fine: it is what it is. Authors would be recommended to better acknowledge these limitations, and not (as they do in the abstract) state they found "20% ... losses in fragmented Amazonian forests" when in fact they only scanned 315 trees. A second minor issues is the grammar and readability, which needs attention before considering publication.

Some minor edits include:

Abstract:

"apace"?

RESULTS/FIGURES

Supp Fig. 5 is confusing, and neither the caption nor corresponding text explain how the red line was calculated.

Supplementary Figure 6. There does not appear to be an explanation for how partitioning of variance was done.

Edge effects on tree allometry

"76" - "76 m"?

Fig. 4 - what is FE+allometric - this is unclear here but better defined in the Disc.

DISCUSSION

Unclear sentence possessing the phrase "at the expense of lower mechanical stability 44". Consider revising.

METHODS

Table 1: Could the first trait be more clearly labeled (e.g. "density"?)

POINT-BY-POINT RESPONSE TO REVIEWERS' COMMENTS

Reviewer #1 (Remarks to the Author):

The study presents new insights into the effects of forest fragmentation on above ground biomass and individual tree architecture in Amazonian forests. The main finding is that forest fragmentation is not only reducing above-ground biomass through edge effects on forest structure, but also by affecting tree architecture and allometry. The authors use terrestrial lidar data to quantify tree architectures and allometry and permanent plot data from long-term experimental sites in Amazonia to quantify above-ground biomass.

The study is interesting from two different perspectives. First, the study addresses a topic that is relevant to understand general patterns of tree architecture in response to changing environmental conditions. Second, the findings have important implications for quantifying biomass and carbon stocks of tropical forests, because the authors show that changes in tree allometry following forest fragmentation may lead to an overestimation of biomass at forest edges, if differences in allometry are not being accounted for in allometric equations. While I find the study very interesting and basically well conducted, there are a few things that remain unclear. Before considering the paper for publication, it would be very helpful if the authors could clarify a few points that I am outlining in the following.

AGB of forest edges and interior:

In figure 4, the authors report that edge effects on tree allometry cause an additional reduction in AGB, which is one of the key findings. I have a few questions and probably some doubts about that finding.

First, I think I do not fully understand the difference between the AGB estimate for forest edges and the forest edges + allometric effects. Is the estimation of AGB for forest edges based on allometric equations that do not take the allometry of trees at the edges into account, while the additional 6.6 Mg ha⁻¹ result from considering differences in allometry?

You are right - the additional AGB loss results from considering differences in allometry caused by edge effects, in addition to edge effects on forest structure during the 40 years of experiment (a combination of tree mortality, tree growth, and recruitment that resulted in changes in tree size, tree density and species composition). We have now made this clear in the methods, results and discussion to avoid confusion regarding the additional edge effects on forest AGB caused by changes in tree architecture. More specifically in the Results section:

“We used allometric models to predict woody volume in both edge and interior forests, unabling us to estimate aboveground biomass (AGB) across larger spatial scales. Linear mixed models applied to data from 44 1-ha permanent plots revealed a statistically significant reduction in AGB of 24.7 Mg ha⁻¹ due to edge effects ($t = -3.1$; P -value = 0.003). This reduction accounted for nearly 10% of the AGB of structurally intact forests (282.2 ± 15.3 Mg ha⁻¹) and comprised two components: first, there was an 18.7 Mg ha⁻¹ decline in AGB due to edge effects on forest structure - caused by fragmentation-related variation in tree size, tree density and species composition - within 100 m from the forest margins; second, there was a 6.0 Mg ha⁻¹ decline in AGB caused by edge effects on tree allometry within 55 m from the forest margins. These distance thresholds were chosen based on previous studies indicating stronger edge effects within 100 m from forest edges on forest structure, and our own study indicating stronger edge effects on tree allometry within 55 m from the forest edges (see Methods for detailed explanations). We visually represented the predicted reduction in AGB caused by edge effects on both forest structure and tree allometry, comparing them to control interior forest plots. Notably, tree allometry alone contributed to one-third of the total AGB decline resulting from edge effects (Figure 4).

Figure 4. Predicted aboveground biomass (AGB) for edge (N = 28 ha) versus interior (N = 16 ha) forest plots within the Biological Dynamics of Forest Fragments Project (BDFFP) in Central Amazonia, the world's longest-running experimental study of habitat fragmentation. Points represent model predictions of AGB in edge versus interior forest plots from linear mixed modelling and error bars denote 95% confidence intervals. 44 1-ha plots used for AGB predictions contained tree measurements on more than 12,000 individual stems ≥ 10 cm diameter across 1026 tree species. The shaded dark grey area corresponds to AGB loss caused by fragmentation-related changes in forest structure owing to edge effects on tree mortality, growth and recruitment. The red area corresponds to AGB loss caused by edge effects on tree allometry, calculated by comparing AGB estimates using an allometric model that considered edge effects on tree allometry with an allometric model developed for interior forests.”

Second, the sample sizes for edge and interior plots strongly differ (30 vs. 6). I think it is crucial to clarify how the plot locations were selected (e.g. randomly vs purposefully selected as representative of the larger area). Even in primary forests, forest structures may spatially vary strongly (e.g. Ehbrecht et al. 2021). With increasing sample size, usually more variability is being captured during sampling. That makes me wonder whether the differences in AGB could just be an artefact resulting from differences in sample size. How do the AGB estimates of edge and interior compare to Amazon-wide estimates of AGB? I think the discussion could benefit from providing a few more references on how edge effects on AGB compare to findings from other studies.

We agree - we have increased the sample size to increase the likelihood of capturing the variability between interior and edge plots. For this, we used other plots near Colosso and Dimona (our TLS sample sites), namely Porto Alegre and Florestal. This increased the number of interior plots from 6 to 16. We also employed linear mixed modelling to analyse AGB values, and included forest site (Colosso, Dimona, Florestal e Porto Alegre) as a random variable to accommodate any unique differences between forest sites that could impact species composition and forest structure, such as soil and topography.

Regarding plot selection, the BDFFP sites are composed of forest fragments originally isolated in 1980-1983 after the conversion of the surrounding mature forest into cattle pastures. The project was designed specifically to investigate the relationship between the size of a forest fragment and its stability and functioning. The experimental design of BDFFP is based on comparisons of a replicated series of forest fragments or reserves of different sizes before and after they were isolated from continuous forests (Bierreghard et al, 1992). This controlled experiment is ideal to measure the edge effects on ecosystem properties. We have included some of these details in the first paragraph of the Methods.

We have also included references in the Discussion that compare our AGB findings with other studies, and how edge effects may affect these values. More specifically: “Variation in the woody volume of trees, coupled with variation in the wood density caused by shifts in species composition, determine the spatial patterns of AGB, which vary considerably across undisturbed Amazonian forests. Intact forests at a site about 15 km south of our study area, in similar lowland forests, were found to have mean AGB values of $325.5 \pm \text{CI}_{95\%} 13.6 \text{ Mg ha}^{-1}$

in comparison to our estimates of $282.2 + 15.3 \text{ Mg ha}^{-1}$ ⁵¹. This demonstrates that our estimates of AGB are conservative. Indeed, AGB estimates can vary significantly among equations, as the choice of allometric models and measurement uncertainty, leading to uncertainties of up to 31% and 16%, respectively, in the AGB estimation of Amazonian forests⁵². Furthermore, local environmental conditions that vary at small spatial scales, such as soil fertility, can account for a third of the variation in AGB in terra-firme Amazonian forests⁵³.

Forests in edges tend to experience a dramatic decrease in AGB caused by changes in forest structure, and are often structurally similar to secondary forests⁵⁴. We observed a reduction of 18.7 Mg ha^{-1} (6.6%) in the AGB of forest fragments owing to changes in forest structure within the first 100 m from the forest edges. This reduction resulted from combined edge effects on tree mortality, growth and recruitment that influence tree size, tree density and species composition. Forest fragmentation not only increases the mortality of large trees^{22,55,56}, but can also lead to higher growth rates of large trees and the recruitment of new individuals⁵⁷. However, large tree mortality represents a significant proportion of biomass loss, contributing to the high carbon emissions of fragmented ecosystems^{22,58,59}. The observed reduction in AGB in our study area near the edges was lower than anticipated, potentially due to the controlled conditions in the BDFFP that minimise additional anthropogenic influences such as illegal logging, hunting, fire penetration and pollution⁶⁰. Indeed, edge effects on forest structure in Amazonian forests can extend up to 1000 m, as fragments are more susceptible to recurring disturbances from fires and illegal logging, and sensitive to the land use in the matrix surrounding the forest fragment⁶¹. Moreover, edge effects can be stronger in the initial decades post-fragmentation, with up to 36% of the forest biomass lost in the first 10 to 17 years after fragmentation, potentially followed by a recovery in the subsequent decades^{22,62}. Long-term monitoring of edge and interior forest plots is crucial to investigating edge effects on AGB dynamics, as confounding environmental factors such as soil and topography, which shape species composition and forest structure, can influence the recovery of forest edges and our ability to quantify it.”

Third, I do not fully understand how the authors define forest edges. In Supplementary Figure 1, the BDFFP study site is shown with different transect length for edge (100 m) and interior (30 m)

forests. Does the length of these transects represent the area of forest edges (in terms of depth/width) used for comparing AGB between interior and edges? In the results section, the authors state that they compared AGB estimates within 55 m from forest margins to predict impacts of edge effects on tree allometry and subsequently AGB. In the next paragraph, they state that within 100 m from forest margins AGB is reduced compared to interior forest. Some of the 100 m transects shown in Supplementary Figure 1 cover the entire forest patch (where is the interior part then?). First, I am wondering why the authors did not use the information from the entire 100 m transect for the allometry part? Second, it remains unclear whether AGB and interior structure in 1 ha and maybe 10 ha fragments are comparable to those of 100 ha fragments, as the interior parts in the small fragments may also already show differences in allometry? Other studies for example found deep reaching edge effects of up to 200 m (see Nguyen et al., 2023, <https://doi.org/10.1007/s10980-023-01609-x>). Overall, the study design descriptions leave me a bit puzzled. I think the paper could benefit from a more straightforward study site and study design description. For example, I would recommend to include a map that shows the location of the plots that were used to estimate AGB for edges and interior and provide a clear definition of forest edge in terms of area and distance from the forest margin.

Thank you for this comment. The text explaining edge effects extent on allometry was unclear. We have revised it to explain it in more detail. More specifically in the Results section:

“We used allometric models to predict woody volume in both edge and interior forests, enabling us to estimate aboveground biomass (AGB) across larger spatial scales. Linear mixed models applied to data from 44 1-ha permanent plots revealed a statistically significant reduction in AGB of 24.7 Mg ha⁻¹ due to edge effects ($t = -3.1$; P -value = 0.003). This reduction accounted for nearly 10% of the AGB of structurally intact forests (282.2 ± 15.3 Mg ha⁻¹) and comprised two components: first, there was an 18.7 Mg ha⁻¹ decline in AGB due to edge effects on forest structure - caused by fragmentation-related variation in tree size, tree density and species composition - within 100 m from the forest margins; second, there was a 6.0 Mg ha⁻¹ decline in AGB caused by edge effects on tree allometry within 55 m from the forest margins. These distance thresholds were chosen based on previous studies indicating stronger edge effects within 100 m from forest edges on forest structure, and our own study indicating stronger edge effects on tree allometry within 55

m from the forest edges (see Methods for detailed explanations). We visually represented the predicted reduction in AGB caused by edge effects on both forest structure and tree allometry, comparing them to control interior forest plots. Notably, tree allometry alone contributed to one-third of the total AGB decline resulting from edge effects (Figure 4).”

We have also included two maps that describe the BDFFP experiment in more detail (Figure 5), and a map that shows where the permanent plots (where field-based tree measurements were done for AGB estimation) and TLS transects were located (Figure 6).

Figure 5. The Biological Dynamics of Forest Fragments Project (BDFFP), the world’s longest-running experimental study of habitat fragmentation, is located in Central Amazonia. The BDFFP sites are composed of forest fragments originally isolated in 1980-1983 after the conversion of the surrounding mature forest into cattle pastures. The forest fragments are surrounded by a 100 m matrix, regularly cleaned by cutting the regrowth vegetation to keep the forest fragment isolated.

Figure 6. Allometric models for woody volume estimates were developed from tree measurements using a Terrestrial Laser Scanner (TLS) within the Biological Dynamics of Forest Fragments Project (BDFFP) in Central Amazonia. Six transects (five of 100 x 10 m at the edges and one of 30 x 10 m at the forest interior), denoted as black lines in the Colosso and Dimona sites, allowed us to obtain high quality point clouds to segment and extract 315 trees. Our findings demonstrate that edge effects on tree allometry penetrated 55 m from the forest edges. Tree measurements of stem size combined with species identification across 44 1-ha permanent plots within Florestal, Dimona, Colosso and Porto Alegre were used to estimate the aboveground biomass (AGB) in edge (yellow plots) and interior (white plots) forest plots. Comparisons of AGB values using edge- versus interior-specific allometric equations were made to predict the influences of tree allometry on the forest AGB.

Tree size thresholds:

In the results section about edge effects on architectural traits, the authors mention that trees above 20 m in height were considered to have established before forest fragmentation and that trees below 20 m were most likely post-fragmentation recruits. In Supplementary Figure 4, the authors show that is a considerable number of trees smaller than 20 m (in 2019) that were first measured in the early 1980's. Doesn't this imply that the threshold is rather arbitrary and that there is likely a not too small proportion of trees that are < 20 m but that recruited before the experiment was established? Wouldn't it make sense to derive such a threshold from tree growth models? How would different thresholds affect the results?

Thank you for your question and suggestions. Considering the high number of species in our study area (over 1000 species in 44 1-ha plots), and that growth trajectories can vary considerably among species, we are not able to derive such thresholds from tree growth models. And this threshold does not impact our results, as we used this threshold only to give a perspective that large trees are those that most probably have survived the 40 years of forest fragmentation. Indeed, $\frac{1}{3}$ of all trees ≤ 20 m in height are those that have survived the fragmentation effects since the fragments were isolated in 1980, and thus correspond to short-stature adult individuals that, analogous to the tall surviving trees, have experienced edge effects in their adult stage of life.

As the fragment isolation occurred in 1980, the tall trees of our dataset are individuals that likely have survived the fragmentation effects, and differences in their architecture with forest fragmentation may reflect the ability of adult plants to acclimate to edge effects. On the other hand, short trees can be included in three groups: 1) $\sim 2/3$ of these trees are less than 30 years old recruits that have colonised the area after the establishment of the forest fragments; 2) surviving trees from understory species at varying ontogenetic stages; and 3) trees from slow-growing species, including upper canopy species at varying ontogenetic stages. Trees in group 1 have already been exposed to edge effects during their juvenile phase. Conversely, changes in the architecture of trees in groups 2 and 3 reflect acclimation of short-stature adult individuals to edge effects that, analogous to the tall surviving trees, change their architecture during the adult phase.

Competition:

In different sections and paragraphs of the text, the authors state that crown asymmetry indicates the capability to avoid competition for light. I just want to highlight that there are also studies showing that crown asymmetry can also be considered a measure of competition intensity and not necessarily as a measure of competition avoidance (Seidel et al. 2011 <https://doi.org/10.1016/j.foreco.2011.03.008>).

Thank you for this reference. We have included in the ecological interpretation of asymmetry the following:

“Tree asymmetry is a result of competitive pressure from neighbouring trees⁸⁰”

Tree heights:

At several points, the authors speak of reductions in tree height in forest edges. That term always left me a bit confused when reading the manuscript. As the authors mention, trees can be shorter near edges because of damages or mortality of upper crown parts. But is this generally the case for the studied trees in the study sites? Speaking of reduction in tree height in edges implies that something has reduced the height, e.g. a damage. But couldn't it be that some trees are simply shorter due to the different environmental conditions near edges? For example, under better light conditions trees might need to invest less into height growth in order to capture light. In case there is a considerable number of trees that are not shorter because of damages/crown mortality, but rather of different growing conditions or simply age, I suggest to rather speak of “lower” tree heights instead of “reduced” or “reduction”.

Thank you for this great comment. This makes sense and we agree with the reviewer. We have added the following to the discussion:

“This resulted in a 30% decline in woody volume for large trees, possibly due to factors including higher microclimatic stress, increased wind speeds, proliferating lianas, and damages that induced mortality of large trees and branch loss^{23,28,47}. These effects may have led to collateral damage to neighbouring trees, resulting in reduced heights near the forest edges^{31,48,49}. Alternatively, trees in the edges may have grown less in height, or edge effects may have favoured shorter individuals or

species acclimated to edge conditions, as the height-to-diameter ratio is a strong determinant of mechanical safety⁵⁰. Consequently, a potential combination of stem breakage from damages caused by neighbouring trees and selective forces favouring shorter trees for a given DBH led to reduced woody volume in large trees. This aligns with the significant negative effects of human-induced disturbances on tree biomass³¹.”

Minor comments:

The meaning of the last sentence of first paragraph of methods section is unclear.

We removed the sentence “All trees that are in permanent plots within the BDFFP project”, which comprised of the last sentence of first paragraph of Methods section.

Summarizing, I think it is a very relevant and interesting study. However, I see some room for improving the manuscript, especially by a clearer description of the study design. Methods sections should be written in a way that a study is reproducible. In the current form, I would find it difficult to reproduce the study due to a lack of clarity. As a reader, I was always jumping back and forth between results, methods, discussion and SI to get how and what the authors did (which left me a bit confused here and there). By streamlining the outline and text the manuscript could be greatly improved I think. I hope my comments and suggestions can help to improve the manuscript.

Thank you for reviewing this manuscript in great detail. We believe that your comments were all carefully addressed, which have improved the paper considerably. We included more details in the Methods section and incorporated them in the Results section to make the text clearer.

Reviewer #2 (Remarks to the Author):

Amazonian forests are becoming more fragmented. This manuscript employs TLS across Amazonian sites to assess edge effects on tree architecture and allometry/biomass. In a nutshell, authors found that biomass increases due to enhanced light capture were more than offset by branch loss due to increased susceptibility to disturbance and damage at forest edges (eg higher wind turbulence) for new (colonizing) and established individuals.

TLS-based studies like this are increasing in prominence and offer direct methods to measure individual tree biomass and associated biomass without destructive harvest. Results are novel, and the implications on large-scale biomass estimation potentially transformative. That said, I think that it is important to recognize some of the studies limitations: especially its limited sample size and extent as well as relative inattention to the role of inter-species trait differences. For the most part, species are lumped and I wonder how limiting factor would affect a wider ranging study. These limitations are part of the nature of doing labor-intensive, fine-scaled TLS scans so that's fine: it is what it is. Authors would be recommended to better acknowledge these limitations, and not (as they do in the abstract) state they found "20% ... losses in fragmented Amazonian forests" when in fact they only scanned 315 trees.

Thank you very much for reviewing this manuscript and for this comment. We agree with you that we did not account for the within-species variability, when in fact part of our findings were potentially due to a shift in species composition. The within-plot variability represented the largest proportion of variability in all architectural traits. We have included this paragraph in our Results section to demonstrate it:

“The within-plot variability, that can be seen as local effects arising from edge effects, as well as species-specific and ontogenetic influences accounted for most of the trait variability (Supplementary Figure 5). The within-plot variability also includes analytical errors, such as those arising from measurement, co-registration, tree extraction and QSM – although we were unable to quantify them.”

We also included in the Discussion a few sentences that acknowledge the importance of species for trait variability and the limitations of our study:

“Despite our efforts to understand architectural variation in Central Amazonian forests with edge effects, the mechanisms allowing species to change their architecture remain elusive. The within-plot variability, that can be seen as local effects arising from edge effects, as well as species-specific and ontogenetic influences accounted for most of architectural variability. These results may help elucidate the generality of our findings in architectural variation across Central Amazonia, although large-scale Amazonian gradients in topography, edaphic properties and climate remain to be tested. Moreover, a shift in species composition caused by edge effects may have partially contributed to our observed changes in tree architecture⁶⁵. We propose that future research on tree architecture should continue to unravel the interactions of the environment with functional diversity within species. TLS-based data, combined with molecular, genetic and physiological processes regulating tree architecture, can help resolve debates concerning the mechanisms by which trees change their architecture^{1,69} and enhance predictions of plant responses to global changes.”

A second minor issues is the grammar and readability, which needs attention before considering publication.

Thank you for this. We have now revised the grammar and readability throughout this manuscript.

Some minor edits include:

Abstract:

"apace"?

We meant “apace” as synonym of “rapidly”. We are no longer using this term though, as we have deleted the first sentences to make the abstract more concise.

RESULTS/FIGURES

Supp Fig. 5 is confusing, and neither the caption nor corresponding text explain how the red line was calculated.

Thank you. We have included a description of how the vertical red lines (strongest edge effects on each architectural variable) was calculated. Please note that the edge effect extent found in these analyses did not affect any further analysis in AGB, as we have done it only to understand to what extent edge effects can affect architectural traits.

“To estimate the extent of edge effects on each architectural trait, we used mixed linear models (LME, Eq. 1) that contained a variable representing the plot category of location near an edge or in the forest fragment interior (edge effects), following Nunes and colleagues³⁸. The model also included a variable that represented tree height (H), considering that architectural traits co-vary with tree size and ontogenetic stage^{17,39}. Edge effects and tree height were treated as additive terms to examine the significance of fragmentation and tree height on the variation of architectural traits. We also included an interaction term edge effects \times tree height, as fragmentation may have different effects on trees of different heights (Eq. 5). To examine the influences of distance from edges on the allometric relationships between woody volume and $\text{DBH}^2 \text{H}$ or woody volume and DBH, we tested how the log transformed variables interacted with edge effects (Eq. 6 and Eq. 7). Nested effects of forest site (Colosso versus Dimona sites), fragment size (1, 10 or 100 ha) and plot (or transect) identity were treated as random variables (μ), allowing us to account for the nested spatial variation in architectural traits and to include any idiosyncratic differences between forest site, fragment size and micro-environmental variation (i.e., topography, soil) between plots.

$$\text{Architectural trait} = \beta_0 + \beta_1 \times (\text{edge effects}) + \beta_2 \times \text{H} + \beta_3 \times (\text{edge effects}) \times \text{H} + \mu_i + \varepsilon_i \quad \text{(Eq. 5)}$$

$$\ln(\text{woody volume}) = \beta_0 + \beta_1 \times (\text{edge effects}) \times \ln(\text{DBH}^2 \text{H}) + \mu_i + \varepsilon_i \quad \text{(Eq. 6)}$$

$$\ln(\text{woody volume}) = \beta_0 + \beta_1 \times (\text{edge effects}) \times \ln(\text{DBH}) + \mu_i + \varepsilon_i \quad \text{(Eq. 7)}$$

where β_0 to β_3 are the model parameters, μ_i is the random intercept for the nested effects of region i , fragment size i and plot identity i , and ε_i is the normally distributed residual error.

We then tested the influence of distance to edges on each architectural trait with distances to edges varying between 1 and 100 m, including tree woody volume. We then tested the influence of distance to edges, varying from 1 to 100 m, on each architectural trait; we determined the edge

effects extent for each architectural trait based on the maximum absolute t-value of the term “edge effects” of the model (Supplementary Figure 4). The edge effects extent was then used to categorise our analysis into edge versus interior trees during all trait analyses.”

Supplementary Figure 6. There does not appear to be an explanation for how partitioning of variance was done.

Thank you for this comment. We have now included the following in the Supplementary Methods 5:

“The spatial position with plot identity nested within landscape, and landscape nested within forest region had effects that varied depending on the architectural metric (Supplementary Figure 6). We examined the explained variance by the random variables of Eq.5 to investigate the spatial variability of architectural traits arising from region, landscape and plot (Supplementary Figure 6). The LME models were fitted using the lme function in the “nlme” R package.”

Edge effects on tree allometry

"76" - "76 m"?

Thank you. We have included 76 m.

Fig. 4 - what is FE+allometric - this is unclear here but better defined in the Disc.

We agree with the reviewer that this was unclear. We have updated the Results with a new Figure 4 and more details on what these changes in AGB represent:

“We used allometric models to predict woody volume in both edge and interior forests, enabling us to estimate aboveground biomass (AGB) across larger spatial scales. Linear mixed models applied to data from 44 1-ha permanent plots revealed a statistically significant reduction in AGB of 24.7 Mg ha⁻¹ due to edge effects ($t = -3.1$; P -value = 0.003). This reduction accounted for nearly 10% of the AGB of structurally intact forests (282.2 ± 15.3 Mg ha⁻¹) and comprised two components: first, there was an 18.7 Mg ha⁻¹ decline in AGB due to edge effects on forest structure

- caused by fragmentation-related variation in tree size, tree density and species composition - within 100 m from the forest margins; second, there was a 6.0 Mg ha⁻¹ decline in AGB caused by edge effects on tree allometry within 55 m from the forest margins. These distance thresholds were chosen based on previous studies indicating stronger edge effects within 100 m from forest edges on forest structure, and our own study indicating stronger edge effects on tree allometry within 55 m from the forest edges (see Methods for detailed explanations). We visually represented the predicted reduction in AGB caused by edge effects on both forest structure and tree allometry, comparing them to control interior forest plots. Notably, tree allometry alone contributed to one-third of the total AGB decline resulting from edge effects (Figure 4).”

“Figure 4. Predicted aboveground biomass (AGB) for edge (N = 28 ha) versus interior (N = 16 ha) forest plots within the Biological Dynamics of Forest Fragments Project (BDFFP) in Central Amazonia, the world's longest-running experimental study of habitat fragmentation. Points represent model predictions of AGB in edge versus interior forest plots from linear mixed modelling and error bars denote 95% confidence intervals. 44 1-ha plots used for AGB predictions contained tree measurements on more than 12,000 individual stems ≥ 10 cm diameter across 1026

tree species. The shaded dark grey area corresponds to AGB loss caused by fragmentation-related changes in forest structure owing to edge effects on tree mortality, growth and recruitment. The red area corresponds to AGB loss caused by edge effects on tree allometry, calculated by comparing AGB estimates using an allometric model that considered edge effects on tree allometry with an allometric model developed for interior forests.”

DISCUSSION

Unclear sentence possessing the phrase "at the expense of lower mechanical stability 44". Consider revising.

Thank you. We have revised it to “this may lead to a lower mechanical stability”.

METHODS

Table 1: Could the first trait be more clearly labeled (e.g. "density"?)

Thank you for your question. The trait trunk or branch area per unit volume represents the surface area per unit volume of wood. This trait has been labeled as “surface area per unit volume” in 1978 by Halle et al. in the book “Tropical Trees and Forests: An Architectural Analysis”. This trait is interesting from two perspectives: first, it is related to how thick the branches and trunks are, as low values represent branches and trunks that are thicker because they have large volume, and thus can be linked to higher support of aerial structures for light capture and productivity. Second, higher surface areas (thinner branches and trunks) indicate higher interaction with the atmosphere, including higher respiration rates of the living tissues. This book was our inspiration to use this trait to potentially understand how edge effects affected architectural traits linked to structure for light capture. We have kept the same name along the manuscript for consistency with the book.

REVIEWERS' COMMENTS

Reviewer #1 (Remarks to the Author):

Dear Nunes et al.,

many thanks for the thorough revision of the manuscript. I really appreciate your detailed feedback on my comments and the additional analyses. I only have a few further minor comments, mainly for clarification:

L 38 – The term “50% more wood” is a bit ambiguous. Do you mean more biomass or more volume (or productivity)?

L 39-41. I think the first and second half of the sentence kind of contradict. Maybe remove the first half (which relates to the previous sentence?) and start with something like “however, a disproportionately lower height in some large trees led to a 30% decline in their woody volume”?

L102-104. I would generally suggest to avoid phrases like “first study” or “first to show...”.

L 319. Why are your estimates conservative? Maybe you could add a “..., because AGB estimates can vary significantly among equations, ...” (if that’s why you think you’re estimates are conservative)

L 415. I think something went wrong during the revision of the manuscript. In the methods section on TLS, the previous paragraph is repeated.

Reviewer #2 (Remarks to the Author):

[Editor's note: the reviewer had no further concerns]

Point-by-point response to the reviewers

Reviewer #1 (Remarks to the Author):

Dear Nunes et al.,

many thanks for the thorough revision of the manuscript. I really appreciate your detailed feedback on my comments and the additional analyses. I only have a few further minor comments, mainly for clarification:

L 38 – The term “50% more wood” is a bit ambiguous. Do you mean more biomass or more volume (or productivity)?

More woody volume. We have included this term in the abstract.

L 39-41. I think the first and second half of the sentence kind of contradict. Maybe remove the first half (which relates to the previous sentence?) and start with something like “however, a disproportionally lower height in some large trees led to a 30% decline in their woody volume”?

Thank you. It does make sense the reviewer’s suggestion – we have removed the first half.

L102-104. I would generally suggest to avoid phrases like “first study” or “first to show...”.

We have removed the sentence that mentioned these terms.

L 319. Why are your estimates conservative? Maybe you could add a “..., because AGB estimates can vary significantly among equations, ...” (if that’s why you think you’re estimates are conservative)

Our estimates are conservative because they are lower than estimates for the same region. And this might arise from variations related to the choice of allometric models and measurement uncertainty, as we explained in the following sentence. We have included this in the Discussion as “This demonstrates that our estimates of AGB are conservative given their lower values relative to other estimates for the same region.”

L 415. I think something went wrong during the revision of the manuscript. In the methods section on TLS, the previous paragraph is repeated.

Thank you very much. We have corrected this.

Reviewer #2 (Remarks to the Author):

[Editor's note: the reviewer had no further concerns]